# Multiplexed Imaging Reveals the Spatial Relationship of the Extracellular Acidity-Targeting pHLIP with Necrosis, Hypoxia, and the Integrin-Targeting cRGD Peptide

**DOI:** 10.3390/cells11213499

**Published:** 2022-11-04

**Authors:** Zhao-Hui Jin, Atsushi B. Tsuji, Mélissa Degardin, Pascal Dumy, Didier Boturyn, Tatsuya Higashi

**Affiliations:** 1Department of Molecular Imaging and Theranostics, Institute for Quantum Medical Science, National Institutes for Quantum Science and Technology, Chiba 263-8555, Japan; 2Département de Chimie Moléculaire, Université Grenoble Alpes, CNRS, 38058 Grenoble, France; 3Institut des Biomolécules Max Mousseron, École Nationale Supérieure de Chimie de Montpellier, Université de Montpellier, 34000 Montpellier, France

**Keywords:** pHLIP, tumor acidity, tumor necrosis, tumor hypoxia, α_V_β_3_ integrin, cRGD peptide, intratumoral distribution, multiplexed imaging, spatial relationship

## Abstract

pH (low) insertion peptides (pHLIPs) have been developed for cancer imaging and therapy targeting the acidic extracellular microenvironment. However, the characteristics of intratumoral distribution (ITD) of pHLIPs are not yet fully understood. This study aimed to reveal the details of the ITD of pHLIPs and their spatial relationship with other tumor features of concern. The fluorescent dye-labeled pHLIPs were intravenously administered to subcutaneous xenograft mouse models of U87MG and IGR-OV1 expressing α_V_β_3_ integrins (using large necrotic tumors). The α_V_β_3_ integrin-targeting Cy5.5-RAFT-c(-RGDfK-)_4_ was used as a reference. In vivo and ex vivo fluorescence imaging, whole-tumor section imaging, fluorescence microscopy, and multiplexed fluorescence colocalization analysis were performed. The ITD of fluorescent dye-labeled pHLIPs was heterogeneous, having a high degree of colocalization with necrosis. A direct one-to-one comparison of highly magnified images revealed the cellular localization of pHLIP in pyknotic, karyorrhexis, and karyolytic necrotic cells. pHLIP and hypoxia were spatially contiguous but not overlapping cellularly. The hypoxic region was found between the ITDs of pHLIP and the cRGD peptide and the Ki-67 proliferative activity remained detectable in the pHLIP-accumulated regions. The results provide a better understanding of the characteristics of ITD of pHLIPs, leading to new insights into the theranostic applications of pHLIPs.

## 1. Introduction

Recent advances in cancer therapy have improved the prognosis of most cancers, but unfortunately, the prognoses of some cancers remain poor [1]. The combined approach of conventional cancer cell-targeted therapy and therapeutic resistance-targeted therapy is expected to improve the prognosis of refractory cancers. Recently, we demonstrated that a combination of two radioagents with complementary intratumoral distribution (ITD) improved the antitumor effect in a tumor mouse model [2]: co-injection of ^64^Cu-cyclam-RAFT-c(-RGDfK-)_4_ (^64^Cu-RaftRGD) targeting α_V_β_3_ integrins (α_V_β_3_) on cancer cells [3,4] and ^64^Cu-diacetyl-bis(*N*^4^-methylthiosemicarbazone) (^64^Cu-ATSM) targeting hypoxic metabolism [5,6] suppressed tumor growth and increased survival in a U87MG glioblastoma mouse model, because of the more uniform ITD of radioactivity [2]. To achieve increased efficacy, more cytotoxic radionuclides, such as α-emitters like ^225^Ac [7], are required. RaftRGD can be labeled with ^225^Ac [8,9], but unfortunately, ATSM cannot (owing to its basic structure) [10]. Therefore, it is necessary to find a new hypoxia-targeted chemical that can be labeled with α-emitting radionuclides.

The pH (low) insertion peptide (pHLIP) is a candidate for hypoxia-targeted agents labeled with α-emitters [11]. The pHLIPs, derived from the C-helix of the protein bacteriorhodopsin and first reported by Hunt et al. in 1997 [12], are a class of pH-sensitive peptides that bind to the cell membrane through insertion and folding across the cellular membrane lipid bilayers when the extracellular environment is acidic [13,14,15,16]. The pHLIP sequences (parent and variants) are primarily composed of three regions: an N-terminal extracellular region (for labeling), a middle transmembrane region (for binding to the cell membrane through insertion and folding), and a C-terminal intracellular region (for cytoplasmic drug delivery) [14,17,18]. Owing to the hallmark ubiquitous feature of an acidic extracellular environment of most solid tumors and related to tumor progression [19,20], pHLIPs have long been developed and proven useful in the experimental targeting of tumor acidity for cancer imaging, fluorescence-guided surgery, and therapeutic drug delivery [11,17,18]. Recently, a radiolabeled pHLIP has entered a phase 1 clinical trial for breast cancer imaging [21].

The tumor-targeting ability of pHLIPs has been widely demonstrated in tumor xenografts or allografts in mice [11,22], spontaneously arising tumors in transgenic mice [23,24,25], and clinical biopsy samples [26,27], involving a variety of cancer types. The pH-dependent tumor-targeting specificity of pHLIPs, labeled with various imaging cargos including fluorescent dyes [13,15,23,25] and radionuclides (^64^Cu, ^18^F, ^99m^Tc, ^68^Ga, or ^89^Zr) [28,29,30,31,32,33,34], has been firmly verified. Tumor-uptake levels of the pHLIPs have been shown to correlate with levels of tumor acidity [30,31]. pHLIPs accumulate in the hypoxic region due to the causes and consequences of hypoxia and acidity in tumors [35,36]. However, there are only a few reports on the spatial colocalization of pHLIPs with poor vascular perfusion [24,29], hypoxia [24,31], and necrosis [31,32] with low resolution. To verify the potential of pHLIP as a hypoxia-targeted therapeutic agent, it is necessary to evaluate how the ITD of pHLIPs correlates with these parameters in detail.

The present study aimed to investigate the ITD of pHLIPs to explore the potential of a pHLIP-based strategy to be applied as an alternative approach to targeting hypoxia and/or be incorporated into our proposed rationale for pairing α_V_β_3_- and hypoxia-targeted radiotherapies. Here, we investigated the detailed ITD of pHLIPs in relation to necrosis, hypoxia, and other pathological features of concern including microvessels, lipid droplets, macrophages, and proliferation, with the α_V_β_3_-targeting cRGD peptide (Cy5.5-RaftRGD), using two tumor models (α_V_β_3_-positive U87MG and IGR-OV1) [4,37]. We chose pHLIP variant 3, a representative member of pHLIPs with optimized tumor targeting [15,26,38], conducted a series of imaging studies from the mouse whole-body to tumor-cell level, and used our unique multiplexed fluorescence colocalization analysis.

## 2. Materials and Methods

### 2.1. Peptide Synthesis and Fluorescent Dye Conjugation

The 27-residue peptide pHLIP variant 3 (ACDDQNPWRAYLDLLFPTDTLLLDLLW) was synthesized and fluorescently labeled based on the methods previously reported [24,38]. Briefly, the peptide was synthesized by the standard Fmoc-based solid-phase peptide synthesis method and the crude product was purified by reverse-phase chromatography. The peptide was then conjugated, via the N-terminal Cys residue, with either the fluorescent dye Alexa Fluor™ 546 C5 (AF546, excitation maximum/emission maximum [ex_max_/em_max_]: 556/573 nm) maleimide (Thermo Fisher Scientific K.K., Tokyo, Japan) or near-infrared (NIR) IRDye^®^ 800CW (IR800, ex_max_/em_max_: 774/800 nm) maleimide (LI-COR, Inc., Lincoln, NE, USA), according to the manufacturer’s instructions and finally, purified as mentioned above. The fluorescent pHLIP constructs, AF546-pHLIP (molecular weight [MW]: 4256.4; purity: >98%; Figure 1A) and IR800-pHLIP (MW: 4345.4; purity: >97%; Figure 1A), were produced and purchased from Scrum Inc. (Tokyo, Japan). The Cy5.5-RaftRGD (MW: 5234.3; purity: >98%; ex_max_/em_max_: 675/694 nm) was synthesized as reported previously [39].

### 2.2. Cells, Animals, and Tumor Models

The human glioblastoma U87MG cell line was purchased from the American Type Culture Collection (Manassas, VA, USA). The human epithelial ovarian adenocarcinoma IGR-OV1 cell line was obtained through a Material Transfer Agreement (#1-5734-18) from the National Cancer Institute Developmental Therapeutics Program Tumor Repository (Frederick, MD, USA). The two cell lines were cultured and maintained as previously described [4,37].

Female athymic BALB/cAJcl-*nu/nu* mice (4–5 weeks old) were purchased from CLEA (Japan, Inc. Tokyo) and bred in a specific pathogen-free facility. Animal procedures were approved (Protocol No. 13-1022-9) by the Animal Ethics Committee of the National Institutes for Quantum Science and Technology (Chiba, Japan) and were conducted according to institutional guidelines. Tumor models were established by subcutaneous injection of 5 × 10^6^ U87MG or IGR-OV1 cells in the right flank of the mice. Subsequent experiments were performed when the tumor diameter reached 10–15 mm (3–4 weeks after xenografting).

### 2.3. In Vivo and Ex Vivo NIR Fluorescence (NIRF) Imaging

The VISQUE™ InVivo Smart-LF imaging system (Vieworks Co., Ltd., Anyang-si, Korea) was used. Tumor (U87MG or IGR-OV1)-bearing mice (*n* = 3–5/group) were anesthetized and imaged prior to and at 10 min, 30 min, 1 h, 2 h, 3 h, 4 h, 6 h, and 24 h after intravenous injection (via the tail vein) of IR800-pHLIP (5 or 10 nmol) [24,38] with an ICG filter set (ex_max_/em_max_: 740–790/810–860 nm) and at the same settings (500 ms exposure, middle light intensity, and low-gain mode).

Ex vivo imaging was additionally performed at 4 h [24,39] after intravenous injection of either IR800-pHLIP (5 nmol) or a cocktail with Cy5.5-RaftRGD (10 nmol) [37]. Thereafter, the mice were euthanized via isoflurane saturation, and the tumors and major organs were removed and imaged in parallel by VISQUE. IR800 fluorescence was visualized as described above, and the Cy5.5 signal was read by changing to a Cy5.5 filter set (ex_max_/em_max_: 630–680/690–740 nm).

Image processing (pseudocoloring and superimposing) and measurements of fluorescence intensity of the region of interest (ROI) were performed using VISQUE. The ROIs were drawn to encompass the tumors, and mean surface intensities were calculated.

### 2.4. Multiplexed Fluorescence Imaging of Tumor Sections and Histology

A cocktail of either AF546-pHLIP + IR800-pHLIP (5 nmol each) or that of IR800-pHLIP + Cy5.5-RaftRGD (5 and 10 nmol, respectively) was administered to the mice and the mice were examined at 4 h post-injection. Pimonidazole, a commonly used exogenous hypoxia marker in histochemistry (1.5 mg, Hypoxyprobe-1 Omni Kit; Hypoxyprobe, Burlington, MA, USA), was administered intravenously to the mice 1 h before euthanizing. Tumors were removed and cryo-sectioned in sequential sections (10 µm thickness) as previously described [40]. Sections were stored in the dark at −80 °C until subsequent experiments.

### 2.5. ITD of pHLIP and Relation to the Tumor Microenvironment

#### 2.5.1. Whole-Tumor Section Imaging

Frozen sections of tumors treated with AF546-pHLIP, IR800-pHLIP, and pimonidazole were air-dried without further processing and scanned at 800 nm using the Odyssey CLx NIR Imaging System (21 µm resolution; LI-COR) to visualize IR800 fluorescence throughout the section. Representative samples were further imaged using the Keyence BZ-X800 fluorescence microscopy system (Keyence, Osaka, Japan) with Texas-Red and Cy7 filter sets to obtain the whole-section images of AF546-pHLIP and IR800-pHLIP, respectively. Sections were then subjected to double staining (acetone fixation) of the microvascular biomarker CD31 and pimonidazole using rat anti-CD31 (1:1500 dilution; BD Pharmingen, San Diego, CA, USA) and rabbit anti-pimonidazole (1:200 dilution; Hypoxyprobe-1 Omni Kit), followed by Alexa Fluor 594–conjugated goat anti-rat IgG (1:200 dilution; Invitrogen, Waltham, MA, USA) and IR800–conjugated goat anti-rabbit IgG (1:100 dilution; LI-COR) as previously described [37]. Nuclei were counterstained with 4′,6-diamidino-2-phenylindole (DAPI) using a slide encapsulant (Dapi-Fluoromount-GTM; SouthernBiotech, Birmingham, AL, USA). Corresponding whole-section images of CD31, pimonidazole, and DAPI were captured using the Keyence with TRITC, Cy7, and DAPI filter sets, respectively. Hematoxylin and eosin (H&E) staining was performed on adjacent sections. Exogenous fluorescence signals were significantly weakened during the immunostaining process, and therefore, did not interfere with the immunofluorescence signals. Similar protocols for tissue processing, staining, and image acquisition were applied for subsequent multiplexed imaging procedures. Sequential images were taken at the same resolution and capture position settings.

#### 2.5.2. High-Resolution Regional Imaging

Based on the whole-section images, ROIs (representing differing patterns of pHLIP distribution) were selected for high-resolution imaging using the remaining adjacent sections. One section (paraformaldehyde (PFA)-fixed and H&E-stained) was used to study the morphology of the ITD of pHLIP, and another (CD31 and pimonidazole double-stained) to clarify the spatial relationship between pHLIP and hypoxia in the context of the microvasculature. The PFA fixation, mentioned below, was preferentially used (when necessary) to better retain the cellular structure and tissue integrity. All corresponding images of the same ROI-defined sites were sequentially captured using a 20× objective lens on the Keyence, as mentioned above.

#### 2.5.3. Cellular Level Analysis

After regional imaging, the pimonidazole and CD31 double-stained sections were subjected to H&E staining (if possible). Regional fluorescence colocalization images of pHLIP, pimonidazole, CD31, and DAPI were produced in parallel with the corresponding images of H&E staining. From these images, three representative ROIs with: (i) negligible accumulation of both pHLIP and pimonidazole, (ii) high accumulation of pHLIP, and (iii) high accumulation of pimonidazole were further magnified to detect the morphological details of pHLIP localization at the cellular level.

### 2.6. Spatial Relationships among pHLIP, Lipid Droplets, and Hypoxia

To investigate the possibility that the ITD of pHLIP was attributed to the targeting of lipid droplets (reportedly found in the necrotic area of cancer [41]), representative regional sites of pHLIP distribution were imaged as described in Section 2.5.2. The sections were fixed with PFA and stained with a specific fluorescent probe, Lipi-Green (2.5 μM; Dojindo LD02, Tokyo, Japan) at 4 °C for 24 h. Corresponding images of fluorescently stained lipid droplets were taken using a GFP filter set. Finally, double staining for CD31 and pimonidazole, and the corresponding image acquisition were performed, as described above. In this case of samples fixed in PFA, higher concentrations of anti-CD31 (1:400 dilution) and anti-pimonidazole (1:100 dilution) antibodies were used.

### 2.7. Spatial Relationship between pHLIP and Macrophages

To investigate the possibility of the ITD of pHLIP being attributed to the targeting of tumor-associated macrophages (important components of the tumor microenvironment [42,43]), representative regional sites of pHLIP distribution were imaged as described in Section 2.5.2. The acetone-fixed sections were then subjected to staining of CD11b (a marker of tissue macrophages) using purified rat anti-mouse CD11b monoclonal antibody (Clone M1/70.15; 1:100 dilution; Bio-Rad Laboratories Inc., Hercules, CA, USA) [43,44], followed by Alexa Fluor 594–conjugated goat anti-rat IgG as mentioned above. Corresponding images were taken as described above.

### 2.8. Assessment of Cellular Proliferation in pHLIP-Accumulated Regions

Representative images of AF546-pHLIP-accumulated regions (20× objective lens) were captured. Subsequently, double staining (PFA fixation) was performed by using rabbit anti-Ki-67 (SP6, 1:300 dilution; Abcam, Cambridge, UK) and anti-CD31. Corresponding images were acquired using Cy7 and TRITC filters for Ki-67 and CD31 staining, respectively.

### 2.9. Spatial Relationships among pHLIP, cRGD Peptide, and Hypoxia

The ITD of IR800-pHLIP and Cy5.5-RaftRGD across the whole sections of tumors was visualized using the Odyssey at 800 and 700 nm, respectively. Regional bright-field and NIRF images (20× objective lens) or tiled images (viewed in a wider field) were captured with the Keyence. The fluorescence of IR800-pHLIP was read as described above, while Cy5.5-RaftRGD was detected using a Cy5.5 filter set. After imaging, the sections were subjected to CD31 and pimonidazole double staining, and the corresponding images were acquired as described above.

### 2.10. Spatial Relationship Analysis

The previously obtained images of the objects (a tumor mass, a whole-tumor section, or a ROI in the same section) were registered and merged using Adobe Photoshop 23.3.2 software. Intensity plot profiles of the fluorescence signals were created and superimposed using ImageJ 1.53t software (National Institutes of Health, Bethesda, MD, USA). The spatial relationship between the imaging components was then visually determined.

### 2.11. Statistical Analysis

Quantitative data are presented as the mean ± standard deviation (SD). Student’s *t*-tests (unpaired and two-tailed) were used for two-group comparisons (KaleidaGraph Version 4.0, Synergy Software, Reading, PA, USA). *p* < 0.05 was considered significant.

## 3. Results

### 3.1. In Vivo and Ex Vivo NIRF Imaging

Figure 1B shows representative chronologic NIRF images of mice bearing either U87MG or IGR-OV1 subcutaneous tumors after the intravenous administration of 5 nmol IR800-pHLIP. Both tumor models could be visualized by the NIRF imaging during the observation period (10 min–24 h post-injection), with the fluorescence intensity peaking at 1 h (1598 ± 125% and 1200 ± 49% of the pre-injection levels for U87MG and IGR-OV1 tumors, respectively; Figure 1C). At 4 h post-injection, the fluorescence signals in nontumor tissues (excluding the kidney) were notably reduced, while tumor fluorescence persisted at high levels of 1178 ± 109% and 948 ± 90% in U87MG and IGR-OV1 tumors, respectively (Figure 1B and C). Compared to IGR-OV1, U87MG tumors showed significantly higher fluorescence accumulation (e.g., *p* < 0.001 at 1 h post-injection; Figure 1C). Ex vivo imaging was additionally performed at 4 h post-injection to verify the results of in vivo studies. As represented in IGR-OV1 tumor-bearing mice (Appendix A), the excised tumor mass exhibited significant and heterogeneous fluorescence accumulation in contrast to background levels of fluorescence signals in the muscle, spleen, intestine, heart, and lung. The liver showed weak and homogeneous fluorescence accumulation, but the most prominent fluorescence signal was observed in the kidneys, indicating major renal excretion of pHLIP. In U87MG tumor-bearing mice, increasing the dose of IR800-Var3 from 5 to 10 nmol significantly increased the fluorescence intensity of the tumor (e.g., *p* < 0.01 at 1 h post-injection; Appendix A), suggesting a dose-dependent response.

### 3.2. General Whole-Tumor Section Imaging

The ITDs of AF546-pHLIP, IR800-pHLIP, hypoxia (pimonidazole staining), microvessels (CD31 staining), and nuclear DAPI staining were examined at 4 h post-injection of the cocktail of AF546-pHLIP and IR800-pHLIP. Figure 2 shows the results obtained from the same whole sections of a representative U87MG or IGR-OV1 tumor. Corresponding H&E staining of adjacent slides displayed multiple areas of necrosis of U87MG (Figure 2A; coagulative necrosis) and IGR-OV1 tumors (Figure 2B; cavity-like liquefactive necrosis). In both tumor models, the ITD of pHLIP labeled with two different dyes appeared identical and heterogeneous throughout the section. Multiple comparisons of whole-section images indicated a close spatial relationship of pHLIP distribution with both hypoxia and necrosis (especially at the periphery of massive necrosis shown in Figure 2A). Similar results were shown in additional whole-section images of the other three U87MG (Appendix A) or IGR-OV1 (Appendix A) tumors. In addition, either AF546-pHLIP or IR800-pHLIP accumulation was also observed at the periphery of whole-tumor sections of both tumor models (Figure 2, Appendix A). This is a phenomenon also exhibited by other pHLIPs [23,29,31,32,45]. Enlarged views (Appendix A) indicate that such accumulation was attributed to the background levels of the probes in the tumor capsule and/or in mouse skin (if contained in the tumor sample). Both are probably inherently acidic tissues.

### 3.3. Spatial Relationship between pHLIP and Necrosis

Figure 3 shows representative regional images of the ITD of AF546-pHLIP and IR800-pHLIP and the corresponding bright-field and H&E staining at the same sites in the same sections of U87MG (Figure 3A–C) or IGR-OV1 (Figure 3D) tumors. The high-resolution fields of view further validated the spatially identical distribution of AF546-pHLIP and IR800-pHLIP, with the latter producing less background fluorescence. The ITD of pHLIP revealed various patterns, such as sparsely distributed spots (Figure 3A), patches of various sizes (Figure 3B,D), and peripheral localization in massive necrosis (Figure 3C). Importantly, the regions with increased pHLIP accumulation, regardless of the pattern, were the corresponding necrotic regions (unless the amorphous and liquefactive mass of necrotic material was in the central areas of massive necrosis) detected in both bright-field and H&E images. These results were further supported by enlarged views of the ITD of AF546-pHLIP and histology of the same whole sections of both tumor models (Appendix A). Additionally, unlike that seen in the U87MG model, the IGR-OV1 model did not show obvious accumulation of pHLIP in the peripheral regions of massive necrosis, possibly due to their inherently different types of macro-necrosis: coagulative necrosis of U87MG tumors vs. cavity-like liquefactive necrosis of IGR-OV1 tumors.

### 3.4. Spatial Relationship between pHLIP and Hypoxia

Figure 4 shows a representative regional image of the ITD of AF546-pHLIP and the corresponding fluorescence images of hypoxia and microvessels at the same sites in the same sections of U87MG (Figure 4A–C) or IGR-OV1 (Figure 4D) tumors. The high-resolution fields of view demonstrated that the regions of pHLIP accumulation were located either within the hypoxic regions (diffuse or central) or in proximity (without sharp boundaries), showing some extent of overlap (Figure 4E). The spatial correlation between the ITD of pHLIP and hypoxia seems to be better understood in the context of microvascular distribution. The extent of pHLIP accumulation within a hypoxic patch tended to be negatively correlated with the corresponding microvascular density. Figure 4A shows less accumulation of pHLIP in a relatively large hypoxic patch with a significant number of microvessels remaining, and vice versa (Figure 4B).

### 3.5. Cellular Localization of pHLIP

Figure 5 shows a representative high-resolution merged image of pHLIP, hypoxia, microvessels, and nuclei (Figure 5A) and the corresponding H&E staining (Figure 5B) from the same site in the same section of the U87MG tumor. Enlarged views of ROI (normoxia, hypoxia, and necrosis) enabled a cytological study of pHLIP localization (Figure 5C–H). Marked accumulation of pHLIP was seen in the necrotic region, as opposed to the absence or less accumulation of pHLIP in the normal and hypoxic regions. Fluorescent pHLIP was found to be localized in necrotic cells at various stages (Figure 5G,H), showing colocalization with the membrane of cells in pyknosis (shrunken nuclei) or with the cellular debris in karyorrhexis (fragmented nuclei) or karyolysis (dissolved nuclei), and appeared to be surrounded by pimonidazole-stained hypoxic cells.

### 3.6. Spatial Relationships among pHLIP, Lipid Droplets, and Hypoxia

Representative images of the ITD of AF546-pHLIP, lipid droplets, and hypoxia obtained from the same sites in the same sections of tumors were compared. In general, as shown in Figure 6 (U87MG tumor), decreased staining of lipid droplets was found in both the pHLIP-accumulated and hypoxic regions. The enlarged view showed the clear cellular localization of lipid droplets in normoxic tumor cells. An exception was found at the edge of a hypoxic region adjoining liquefactive necrosis of the IGR-OV1 tumor section (Appendix A), in which increased accumulation was observed for both lipid droplets and pHLIP, with no signs of overlap.

### 3.7. Spatial Relationship between pHLIP and Macrophages

Representative images of the ITD of AF546-pHLIP and CD11b staining (macrophages and activated lymphocytes) obtained from the same sites in the same sections of tumors were compared. In general, as shown in Figure 7 (U87MG tumor) and other sites of U87MG and IGR-OV1 tumors (Appendix A), the increased accumulation of macrophages (CD11b-positive cells) was found in the pHLIP-accumulated region, with no signs of overlap. This indicates that macrophages were not the site of pHLIP localization.

### 3.8. Proliferation Status in pHLIP Accumulated Region

Figure 8 shows a representative image of a region of high pHLIP accumulation with the corresponding Ki-67 (a cellular marker for proliferation), CD31, nuclear staining, and merged images at the same site in the same section of the U87MG tumor. Ki-67-positive staining (DAPI/Ki-67 overlap) of tumor cells was detected not only in the highly vascularized normoxic region but also in the high pHLIP-accumulation region with poor vasculature. This result indicates that some tumor cells (probably hypoxic cells) in the pHLIP-accumulated region maintained some proliferative activity, indicating that this region remains an important therapeutic target.

### 3.9. Spatial Relationships among pHLIP, cRGD Peptide, and Hypoxia

The ITD of IR800-pHLIP, Cy5.5-RaftRGD, hypoxia, microvessels, and nuclear DAPI staining was examined at 4 h post-injection of the cocktail of IR800-pHLIP and Cy5.5-RaftRGD. Figure 9A shows representative NIRF images of IR800-pHLIP, Cy5.5-RaftRGD, and hypoxia in the same whole section of the U87MG tumor. They all showed a heterogeneous distribution pattern throughout the section. The pHLIP and hypoxic regions were found to correspond well with the cRGD nonbinding region. Figure 9B,C show multiple high-resolution fluorescence images and a further magnified and merged image of a ROI indicated in Figure 9A, respectively. The hypoxic region was located between the highly vascularized cRGD-binding region and the poorly vascularized pHLIP-accumulation regions (Figure 9C). Appendix A shows ex vivo NIRF images of IR800-pHLIP and Cy5.5-RaftRGD in excised tumors and major organs of mice with U87MG or IGR-OV1 tumors. The distribution of both peptide probes observed on the cut surface of the bisected tumor was heterogeneous and had spatially different but complementary patterns. Similar findings were also observed in microscopic images of the wide field of view (3.2 mm × 3.2 mm) of the tumor sections (Appendix A).

## 4. Discussion

Extracellular acidity-targeting pHLIPs are under development for cancer imaging and therapy. An in-depth investigation of the ITD of pHLIPs is of paramount significance for providing basic information toward understanding therapeutic limitations and facilitating a rational combination therapy design. Owing to the lack of a histological biomarker for extracellular acidity, studies on the ITD of pHLIPs have so far focused on comparisons with other related features of the tumor microenvironment. If the relationships are to be fully comprehended and unambiguously described, it is imperative that a histological comparison of multiple parameters is performed based on the same whole section, rather than preferentially selected small regions of adjacent sections [37,46]. To the best of our knowledge, the present study is the first to conduct a detailed investigation of the ITD of pHLIP in relation to necrosis, hypoxia, an α_V_β_3_-targeting cRGD peptide, and others of concern by conducting a series of imaging from the mouse whole-body to tumor-cell level, using our unique high-resolution multiplexed fluorescence colocalization analysis. This study unambiguously demonstrated the ITD patterns of pHLIPs and their spatial relationship with necrosis, hypoxia, and the cRGD peptide. A pHLIP-based anticancer agent is a promising candidate for a combination of cancer cell-targeted therapy and hypoxia-targeted therapy.

For this study, it was necessary to analyze tumors that had developed hypoxia and necrosis for optimally comparing the ITD of pHLIP; hence, large-sized tumors (10–15 mm in diameter) were studied. We validated the tumor-targeting ability of IR800-pHLIP by in vivo and ex vivo NIRF imaging in two different human tumor models (U87MG and IGR-OV1; Figure 1B,C, Appendix A), consistent with that shown by Adochite et al. in a mouse mammary tumor model (4T1 allograft) [38]. A difference in the tumor kinetics was noticed. In our case, the tumor fluorescence intensity peaked at 1 h post-injection, faster than the time of 4 h shown in their study. This might be due to the faster blood clearance of IR800-pHLIP with relatively lower levels of tumor uptake in our experimental settings. One possible reason for this could be due to the tumor model. As mentioned, the α_V_β_3_-positive necrotic xenografts (typically nonmetastatic) were selected for the current study because we focused attention on the detailed ITD of pHLIPs associated with necrosis, hypoxia, and integrins. On the other hand, some studies employed tumor models with more aggressive phenotypes (more acidic), such as metastatic xenografts, allografts, and spontaneously developed tumors in transgenic mice [23,24,32,45]. Such aggressive models showed greater tumor uptake of pHLIP-based probes in comparison with nonmetastatic xenografted tumor models including ours (thought to be less acidic) [23,32]. In the present models, living cancer cells were observed around pHLIP-accumulating areas (Figure 3 and Figure 5, Appendix A). This means that pHLIP-based therapeutic agents would target living cancer cells. Further systematic comparative studies could validate this. Nonetheless, the following examination of the ITD of pHLIP was determined at 4 h post-injection, at which time the background fluorescence signals (nontumor tissues apart from the kidneys) decreased markedly while the tumor fluorescence persisted at high levels.

For multiplexed microscopic imaging, pHLIP was also labeled with AF546, a bright dye commonly used for fluorescence microscopy. The simultaneous use of pHLIP labeled with different dyes also helped identify some irrelevant signals (such as autofluorescence). An identical ITD was observed between AF546-pHLIP and IR800-pHLIP at both the whole-section and enlarged area level (Figure 2 and Figure 3). Furthermore, our multiplexed microscopic comparisons were performed on the same whole sections and at the same sites in the same sections at identical resolutions, allowing for precise correlation analyses.

The ITDs of pHLIPs in necrotic U87MG and IGR-OV1 xenografts were heterogeneous throughout the tumor section (Figure 2 and Figure 9A, Appendix A), consistent with other reports on different pHLIP members with various labeling and tumor models [22,23,24,29,31,32,34,45]. Compared to these studies, we have further shown various patterns of pHLIP accumulation that can be detected within a tumor (Figure 3).

Some efforts have been made to clarify the spatial relationship of pHLIPs with necrosis and/or hypoxia [24,31,32]. However, the main limitations of these studies are that the comparisons were made between autoradiography and histology (with intrinsically different levels of resolution) [31,32] or on adjacent sections with section-to-section variation (no matter the types of images) [24,31]. The results, therefore, are suggestive or show a trend but are not yet conclusive. Here, armed with the capability of high-resolution multiplexed fluorescence colocalization analysis performed on the same tumor sections, we identified that the regions of increased accumulation of pHLIP perfectly coincided with the necrotic regions (unless the amorphous and liquefactive mass of necrotic material in the central areas was of massive necrosis; Figure 3, Appendix A). Moreover, when observed at the tissue level (Figure 2, Figure 4 and Figure 5A), the ITD of pHLIP and hypoxia had a spatially contiguous correlation with some extent of overlap. Notably, a direct one-to-one comparison of highly magnified images revealed the cellular localization of pHLIP in necrotic cells of pyknosis, karyorrhexis, and karyolysis (Figure 5G). The pHLIP signals were found within the hypoxic regions and appeared to be surrounded by hypoxic cells (Figure 5E–H). These results could provide a new interpretation in pHLIP-based imaging or novel insights into the development of pHLIP-based therapeutics.

The mechanism of action of pHLIPs—targeting cancer cells by exploiting the extracellular acidic microenvironment—is well understood [13,16,47,48]. Reshetnyak et al. [23] demonstrated that fluorescent (Alexa750) pHLIP was distributed in the extracellular space and cellular membranes of tumor cells in small M4A4 metastatic nodules in mouse lungs and spontaneous prostate tumors in TRAMP mice. Similar results regarding pHLIP distribution were reported by Adochite et al. using submillimeter mouse 4T1 lung metastatic lesions [38] and by Rohani et al. using an MMTV-PyMT breast cancer mouse model [45]. Regarding the issue of necrosis, little information was provided in these studies (focused on non-necrotic tumors). However, when nonmetastatic tumor xenografts with necrotic morphology were used, the findings of the present study (using U87MG and IGR-OV1), together with the supporting results of previous studies with either fluorescent pHLIPs (Hela xenograft model) [23] or radiolabeled pHLIPs (PC3 and LNCaP prostate tumor xenografts) [31,32], have unambiguously demonstrated that the hypoxic regions, including necrotic cancer cells, are the prominent accumulation sites of pHLIPs (possibly due to the moderative accumulation of pHLIPs in the less acidic normoxic regions). Since pHLIPs are lipid bilayer-binding peptides, it was reasonably speculated by the authors and Reshetnyak et al. [23] that this class of peptides might bind to lipids present in necrotic tissues. Indeed, Zoula et al. reported an increased accumulation of lipid droplets principally in the necrotic tissue in a C6 rat brain glioma model [41]. However, as examined and shown in Figure 6 and Appendix A, the possibility of lipid droplets involved in the accumulation of pHLIP in the necrotic region can now be ruled out. On the other hand, the macrophages accumulated in necrotic regions (not surprisingly recruited to clean up necrotic corpses [49,50]). The same areas included many signals of pHLIP although the two were not wholly matched (Figure 7 and Appendix A). Moreover, unlike pHLIP, neither co-injected pimonidazole nor Cy5.5-RaftRGD showed a noticeable accumulation in the necrotic regions (as shown in Figure 9). The localization of pHLIPs was found evident in “fresh, ongoing necrotic patches” rather than the central areas consisting of amorphous or liquefactive masses of necrotic material (e.g., Appendix A). Therefore, the most likely reason why pHLIPs preferentially accumulated in necrotic regions is that pHLIPs could specifically bind to certain products of necrosis that remain in the damaged cell membranes or are released into the extracellular space, rather than simply because necrotic tissues uptake everything very readily. Further mechanism studies would provide new insight into pHLIP-based theranostic strategy in future work.

The spatial relationship between pHLIP and hypoxia is easier to understand in terms of the corresponding microvasculature (Figure 2 and Figure 4). Our results clearly show that the distribution of microvessels affects hypoxia and results in changes in pHLIP accumulation. Less pHLIP accumulation was seen in the early or mildly hypoxic region (still vascularized to some extent, Figure 4A) compared to the marked accumulation of pHLIP in the typically poorly vascularized hypoxic region (Figure 4B–D).

Given that hypoxia is associated with increased tumor aggressiveness and poor prognosis and that hypoxic tumor cells are resistant to treatment, this common feature of solid tumors represents a compelling therapeutic target [51,52,53]. Various strategies targeting hypoxia have been reported. However, limited drug penetration into poorly vascularized hypoxic regions of tumors remains a challenge [54,55]. We propose that a pHLIP-based strategy is a potential alternative to managing hypoxia, based on the following facts: (i) pHLIPs could pass through hypoxic regions and accumulate in necrotic regions; (ii) the ITD of pHLIPs showed a spatially contiguous correlation with hypoxia; and (iii) pHLIPs can be potentially labeled with the highly cytotoxic α-emitter ^225^Ac (potentially overcoming radiation resistance) since pHLIPs have been reported to be labeled with a range of radiometals (^64^Cu, ^99m^Tc, ^68^Ga, or ^89^Zr) via peptide modification [28,30,31,32,33,34].

It should be noted that when developing pHLIP-based radiotherapeutics armed with ^225^Ac, it is also imperative to lower the risk of nephrotoxicity, as pHLIPs can accumulate highly in the kidney (as shown in Figure 1 and Appendix A) and the kidney is the dose-limiting organ of radiotherapy. Recent studies demonstrate that through further modifications and optimization through structure-activity relationship analyses, the lead pHLIP derivatives such as ^64^Cu-labeled NO2A-Cys-Var3 [32] and ^89^Zr-labeled DFO-Cys-Var3 [34] have been successfully developed, showing increasing accumulation in the tumor with a concomitant clearance from nontarget tissues including the kidney. So, it would be the case with ^225^Ac-labeled pHLIPs. Other approaches (reducing renal reabsorption) have been introduced to prevent renal damage related to peptide-based radiotherapy. We previously showed that the co-injection of Gelofusine (succinylated gelatin) and _L_-lysine (positively charged basic amino acid) with ^64^Cu-RaftRGD caused a 30–50% reduction in mouse renal radioactivity levels [56]. In patients treated with ^225^Ac-DOTATOC for somatostatin-receptor-expressing cancers, the amino acid mixture of lysine and arginine formulated in gelatin solution was co-administered as a renal protective cocktail [57].

The current study is the first to demonstrate the spatial relationships between pHLIP, cRGD peptides, and hypoxia within the same tumor. Given the intermediate position of hypoxia between the ITD of pHLIP (in necrotic regions) and a cRGD peptide (in normoxic regions) along with a spatially complementary relationship between the two peptides (Figure 9, Appendix A), we would like to propose a new combination (cRGD + pHLIP) approach that corresponds to our previously proposed rationale for pairing α_V_β_3_- and hypoxia-targeted radiotherapies [2,37]. As mentioned, the proof-of-concept study using the co-injection of ^64^Cu-RaftRGD targeting α_V_β_3_ on cancer cells [3,4] and ^64^Cu-ATSM targeting hypoxic metabolism [5,6] has demonstrated that a combination of two radioagents with complementary ITDs can improve the antitumor effect owing to the increased uniform ITD of radioactivity [2]. Therefore, it is reasonably anticipated that with the combined use of cRGD and pHLIP peptides and both armed with an α-emitter (like ^225^Ac), in addition to producing a uniform ITD of radioactivity, hypoxic tumor cells and/or cancer stem cells (possibly residue within necrotic regions [58,59]) would be effectively killed via the crossfire effect of irradiation delivered from the two radiopeptides in opposite directions. Although a few proliferative tumor cells in pHLIP-accumulated regions (Figure 8) could limit the efficacy of pHLIP-targeted therapy, such combination therapy may also be beneficial due to pHLIP-based elimination of microenvironmental sheltering from cRGD-based monotherapy.

In addition to the previously discussed limitation of this work, there are at least three more notable limitations. First, this work focused on only one member of the pHLIP family (variant 3), although this is a representative member and is reported more frequently in the literature. Second, the ITD characteristics of pHLIPs were investigated in a small number of tumor types (two xenograft models), thus requiring further studies using more tumor types, including more aggressive phenotypes such as allografted tumors. Nevertheless, we hope that researchers in the field will use the study protocols established herein to identify the ITD characteristics of newly developed pHLIP-based drugs toward future clinical translation. The third limitation is that pHLIP compounds have also been reported to target the tumor–stroma interface [34,45], but such a finding was not observed in our experimental settings (Appendix A) with the exclusion of targeting of the macrophages (Figure 7, Appendix A), and it is necessary to further investigate whether they are tumor type dependent.

## 5. Conclusions

We performed a comprehensive investigation (whole-body, excised tumor, whole-tumor section, and microregional sites of tissue and cells) with two tumor mouse models, unambiguously revealing the details of the ITD of a representative pHLIP (variant 3). The spatial relationship of the ITD of pHLIP with multiple parameters was clarified in a straightforward manner using a unique high-resolution multiplexed fluorescence colocalization analysis based on the same whole tumor sections. The ITD of pHLIP was heterogeneous, with increased accumulation in the regions of necrosis by localizing to various morphological patterns of necrotic cells. There was a spatially contiguous but nonoverlapping relationship (at least at the cellular level) between the ITD of pHLIP and hypoxia. Notably, an intermediate position of hypoxia was revealed between the ITD of pHLIP and a cRGD peptide. These results provide a better understanding of the ITD characteristics of pHLIPs, which may lead to new insights into the theranostic applications of pHLIPs. Based on the close spatial relationship of hypoxia with pHLIP and cRGD peptides, we propose an alternative approach, a new “pHLIP + cRGD” combination for tackling treatment-resistant hypoxia via a potential crossfire of the two peptide-based therapeutics.

## Figures and Tables

**Figure 1 cells-11-03499-f001:**
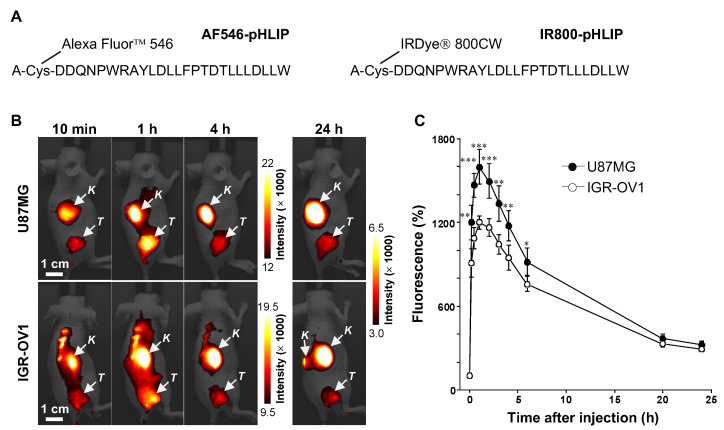
In vivo near-infrared fluorescence (NIRF) imaging. (**A**) The amino acid sequence of pHLIP and fluorescence labeling at the N-terminal side. AF546-pHLIP, Alexa Fluor™ 546 fluorescent dye-conjugated pHLIP; IR800-pHLIP, IRDye^®^ 800CW NIR fluorescent dye-conjugated pHLIP. (**B**,**C**) NIRF imaging of mice bearing U87MG or IGR-OV1 subcutaneous tumors using IR800-pHLIP (5 nmol, intravenous injection). The images were acquired pre- and at 10 min, 30 min, 1 h, 2 h, 3 h, 4 h, 6 h, and 24 h post-injection. (**B**) Representative images. *K*, kidney; *T*, tumor. (**C**) Tumor fluorescence intensity over time. Fluorescence (%), percentage of the pre-injection signal intensity. Data are expressed as mean ± standard deviation (*n* = 5/group). * *p* < 0.05, ** *p* < 0.01, and *** *p* < 0.001 vs. IGR-OV1.

**Figure 2 cells-11-03499-f002:**
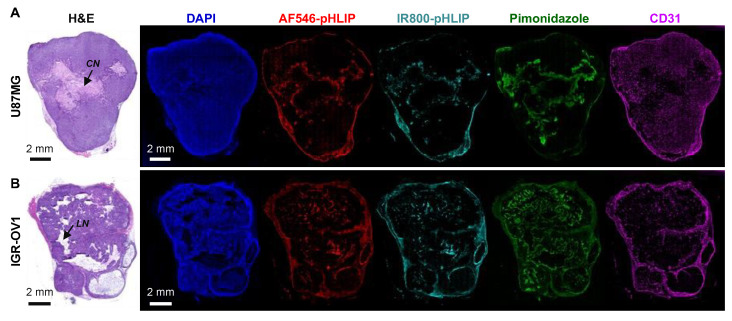
Distribution of pHLIP on whole-tumor sections. U87MG (**A**) or IGR-OV1 (**B**) tumor-bearing mice were intravenously injected with a cocktail of AF546-pHLIP and IR800-pHLIP (5 nmol each) and examined at 4 h post-injection. The images of pHLIP distribution (AF546-pHLIP, red; IR800-pHLIP, aqua) are displayed in parallel with those of histologic sections stained with hematoxylin and eosin (H&E), pimonidazole-stained hypoxia (green), CD31-stained microvessels (magenta), and DAPI-stained nuclei (blue). All fluorescence images were acquired on the same whole sections; H&E staining was performed on adjacent sections. *CN*, representing a region of coagulative necrosis; *LN*, representing a region of liquefactive necrosis (cavity-like).

**Figure 3 cells-11-03499-f003:**
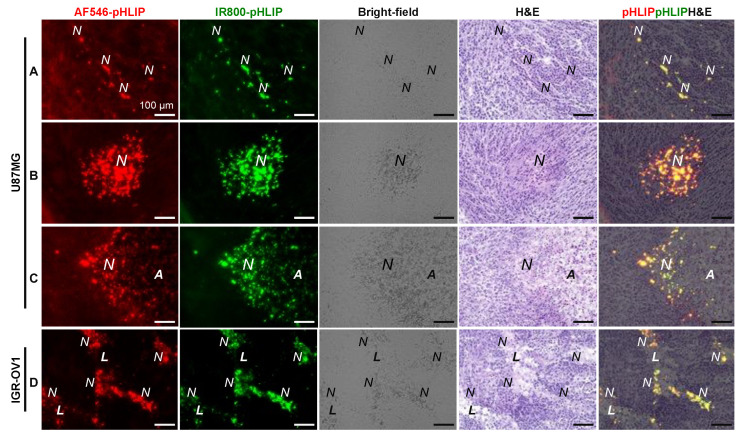
The intratumoral distribution (ITD) of pHLIP in different sites of U87MG (**A**–**C**) or IGR-OV1 tumors (**D**). The images of pHLIP distribution (AF546-pHLIP, red; IR800-pHLIP, green) are displayed in parallel with the corresponding bright-field images and hematoxylin and eosin-stained specimens at the same sites in the same sections. Yellow color in the rightmost panels (**A**–**D**), red/green overlapping. *N*, necrotic region; *A*, amorphous mass of necrotic material; *L*, liquefactive mass of necrotic material.

**Figure 4 cells-11-03499-f004:**
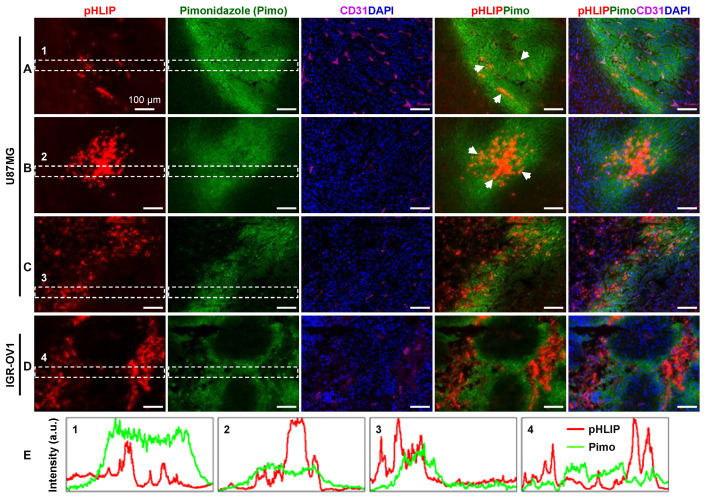
Spatial relationship between the intratumoral distribution (ITD) of pHLIP and hypoxia. Colocalization analysis of pHLIP distribution (AF546-pHLIP, red) and pimonidazole-stained hypoxia (green) was performed at three different sites of a U87MG tumor section (sites (**A**–**C**)) and a representative site of an IGR-OV1 tumor section (**D**), with the corresponding distribution of CD31-stained microvessels (magenta) as a reference. Blue, nuclei stained with DAPI. The red/green overlapping (yellow) is not noticeable, but some orange-colored overlaps are visible (arrowheads). (**E**) Intensity plot profiles of the fluorescence signals in region-1, -2, -3, and -4 marked in (**A**–**D**), respectively.

**Figure 5 cells-11-03499-f005:**
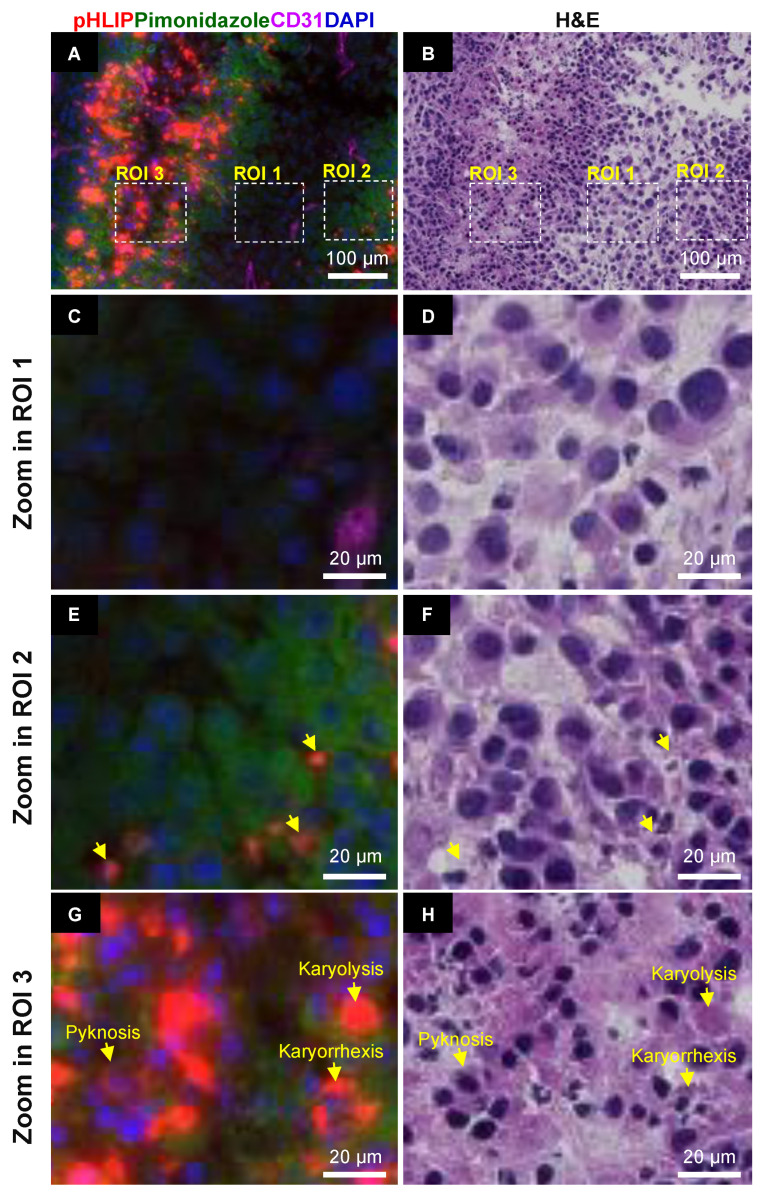
pHLIP localization at the cellular level. (**A**) An overlay image of pHLIP distribution (AF546-pHLIP, red), pimonidazole-stained hypoxia (green), CD31-stained microvessels (magenta), and DAPI-stained nuclei (blue). (**B**) The corresponding hematoxylin and eosin staining in the same site of the same section of a U87MG tumor. (**C**,**E**,**G**) denote enlarged views of ROI-1, -2, and -3 of (**A**), respectively. (**D**,**F**,**H**) depict enlarged views of ROI-1, -2, and -3 of (**B**), respectively. ROI 1, normoxic region; ROI 2, hypoxic region; ROI 3, necrotic region. Yellow arrows in (**E**,**F**) mark the pHLIP signals in the extracellular space. Pyknosis, karyorrhexis, and karyolysis in (**G**,**H**) clearly show the necrotic cells with shrunken, fragmented, and dissolved nuclei, respectively.

**Figure 6 cells-11-03499-f006:**
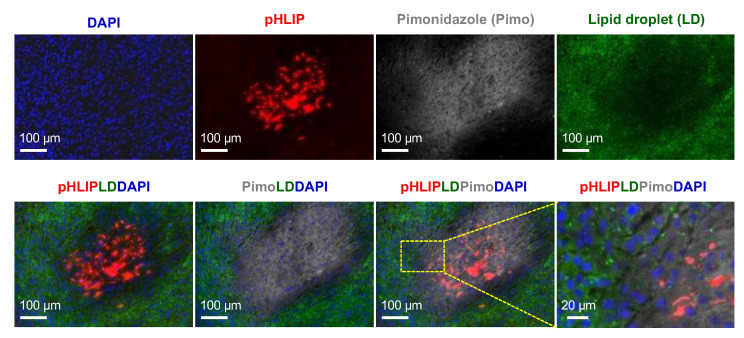
Spatial relationships among the intratumoral distribution (ITD) of pHLIP, hypoxia, and lipid droplets. pHLIP distribution (AF546-pHLIP, red), pimonidazole-stained hypoxia (gray), and lipid droplet staining (green) were examined at the same sites of the same tumor sections. Shown here are images acquired from a representative high pHLIP-accumulated region in a U87MG tumor section. The lipid droplet-specific fluorescent probe Lipi-Green was used for staining the lipid droplets. Blue indicates nuclei stained with DAPI. An enlarged view of the dotted rectangle clearly shows the cellular localization of the lipid droplets.

**Figure 7 cells-11-03499-f007:**
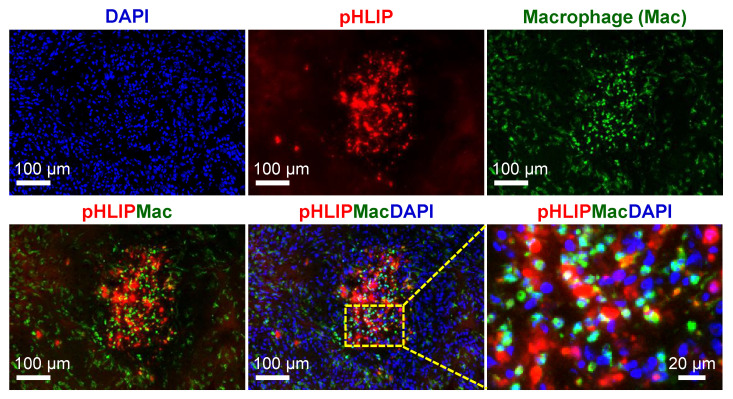
Spatial relationship between the intratumoral distribution (ITD) of pHLIP and macrophages. pHLIP distribution (AF546-pHLIP, red), CD11b-stained macrophages (green), and DAPI-stained nuclei (blue) were examined at the same sites of the same tumor sections. Shown here are images acquired from a representative high pHLIP-accumulated region in a U87MG tumor section. The red/green overlapping (yellow) is not indicated. An enlarged view of the dotted rectangle clearly shows that pHLIPs and macrophages (CD11b-positive cells) were not overlapping.

**Figure 8 cells-11-03499-f008:**
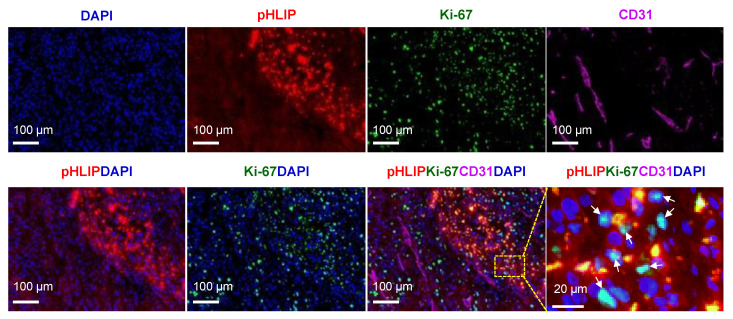
Tumor proliferation status in the pHLIP-accumulated region. pHLIP (AF546-pHLIP) distribution and proliferation status were examined at the same sites of the same tumor sections. Shown here is a representative high pHLIP-accumulated region (red) with double immunofluorescence staining for the proliferation marker Ki-67 (green) and microvessels (CD31, magenta) in a U87MG tumor section. Blue indicates nuclei stained with DAPI. An enlarged view of the dotted rectangle clearly shows the Ki-67-positive stained cells (arrowheads, aqua).

**Figure 9 cells-11-03499-f009:**
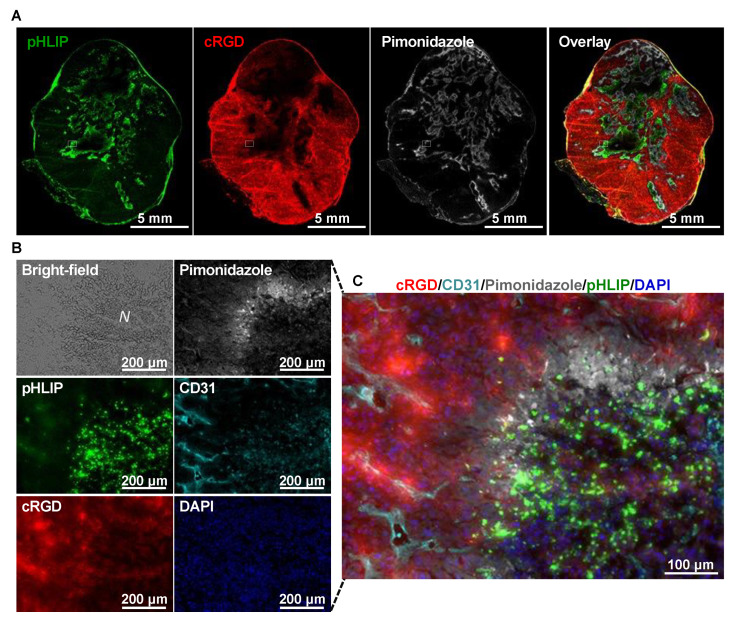
Spatial relationship of the ITD of pHLIP with the cRGD peptide-based probe, Cy5.5-RaftRGD. Tumors were examined at 4 h post-injection of 5 nmol IR800-pHLIP and 10 nmol Cy5.5-RaftRGD. (**A**) Near-infrared fluorescence (NIRF) images of the distribution of pHLIP (green), cRGD (red), and pimonidazole-stained hypoxia (white) on the representative whole section of a U87MG tumor. (**B**) Enlarged views of the same ROI (rectangles indicated in (**A**)) and the corresponding images of bright-field, CD31-stained microvessels (aqua), and DAPI-stained nuclei (blue). *N*, necrotic region. (**C**) Highly magnified fluorescence overlay.

## Data Availability

The data that support the findings of this study are available from the corresponding author upon reasonable request.

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
