# Peer review of "Multiplexed Imaging Reveals the Spatial Relationship of the Extracellular Acidity-Targeting pHLIP with Necrosis, Hypoxia, and the Integrin-Targeting cRGD Peptide"

_cells, 2022, doi:10.3390/cells11213499_

Round 1

Reviewer 1 Report

There are several aspects, which do not correlate with previously published results including: i) decrease of signal in tumor with time (there is a significant amount of data indicating on accumulation of the signal in tumors); ii) difference in tumor distribution compared to the data presented by others (Rohani et al, for example, and others). Authors should provide explanation why their data are different from previously published results.

It is difficult to make conclusion that pHLIP targets necrotic core from the presented images, actually quite an opposite! The fact that pHLIP could be found next to the necrotic areas could be attributed to targeting of TAMs, which are expected to be next to the necrotic areas. Recently it was published report clearly indicating targeting of activated macrophages. Authors should perform IHC for immune cells and stroma, which is very well targeted by pHLIP. It does not make any sense for pHLIP to target necrotic cells, especially when authors ruled out potential targeting of lipid droplets. Thus, targeting of necrotic cells could be accomplished (according to authors logic) due to low pH targeting and it makes no much sense, since necrotic/apoptotic cells are metabolically non-active and should not be producing acidity. Moreover, there are reports that necrotic core has high pH. Any acidity in necrotic areas is expected to come from action of activated macrophages and they can take pHLIP very well.

It is not very clear why different fixative methods are used. Acetone fixation should be avoided.

It is not clear what p-levels are indicating on Fig 1.

Major revision would be required to perform staining and co-localization with immune and stroma cells with TME before reaching to any conclusions.

Author Response

Please find the attached PDF file.

Reviewer 2 Report

The manuscript by ZHao-Hui Jin describes analyses of the spatial distribution of extra-cellular acidity-targeting peptide pHLIP. The addition of pHLIP to RAFT peptide in the context of ligand-based radio-therapy offers the promise to cover non-overlapping regions of the tumors, thus boosting the overall efficiency. The paper is well written and the quality of the presentation is very good. Regrettably, authors rely entirely on eyeballing for their analysis. The use of spatial statistics would have enabled reaching more robust conclusions, and making more nuanced inferences. Still, given the absence of well-established "off the shelf" pipelines for spatial analyses, the extent of co-localization and use of sections across whole tumors rather than "representative images", "eyeballing" appears to be sufficient to support the major inferences. Therefore, I think that the paper is suitable for publication with a minor revision. A more nuanced presentation of some of the conclusions is warranted. Specifically:
  1. The relationship between pHLIP staining and necrosis appears to be more nuanced that stated by the authors. From the data shown in Fig2 and S2-3 there seems to be a meaningful difference between the 2 models, where pHLIP is localized at the boarder of macro-necrotic regions in U87MG border. In IGR-OV1 model pHLIP is mostly missing in the borders of the large necrotic regions. In both models, there seems to be a strong co-localization in borders of fresh, ongoing necrosis. It would have been useful to try to dissect potential reasons for this difference and cover the heterogeneity of necrotic regions in the discussion.
  2. How many tumors were analyzed for whole-tumor section imaging reported in Figure 2? Without quantitation, it is difficult to evaluate variability in co-localization. However, authors should at least include additional images of whole-tumors, to enable evaluation by eyeballing.
  3. Both peptides show strong co-localization at the stromal boarder of the tumors, within well-vascularized regions. Authors should at least comment on it.
  4. I suggest to start presentation of cellular localization, as well as co-localization with lipid droplets and KI-67 with brief statements introducing the rationale for these analyses.
  5. In contrast to the association with necrosis, inferences on co-localization with lipids droplets and KI-67 (and, to a lesser extent, with hypoxia), as well as analyses of cell localization still rely on use of representative images. Given the lack of quantitative analyses, it is hard to evaluate robustness of these inferences. While authors do show that some of the KI-67 positive cells co-localize with pHLIP, the vast majority of proliferation occurs within regions with higher oxygen and nutrients access. Since the main point of the paper is to demonstrate non-overlapping localization between the 2 peptides, this is not a fatal flaw. But it needs to be at least acknowledged.
  6. Given that hypoxia is associated with reduced sensitivity to radiotherapy, and that proliferation within hypoxic regions is limited, it would be useful to at least discuss how this could reduce the potential benefit of the "cross-fire" due to the inclusion of pHLIP in the therapy, especially considering the increased cost and toxicity of combinations. On the other hand, combination therapy might still be beneficial due to the elimination of microenvironmental sheltering from the RAFT-based therapy.

Author Response

Please find the attached PDF file.

Reviewer 3 Report

The current manuscript described their results of characterized distribution of pHLIPs, especially the spatial relationships with hypoxia and necrosis. The experimental data supported their conclusions, could be useful and interesting to the readers who are working on similar topics.

The manuscript was well organized and written. I suggest to publish asap in this journal.

Author Response

Reviewer #3:
General Comments:
The current manuscript described their results of characterized distribution of pHLIPs, especially the spatial relationships with hypoxia and necrosis. The experimental data supported their conclusions, could be useful and interesting to the readers who are working on similar topics.

The manuscript was well organized and written. I suggest to publish asap in this journal.

Authors’ response: We thank the reviewer for their positive evaluation of our work and for recognizing the importance of this manuscript to the field.

Round 2

Reviewer 1 Report

This study contradicts many other published studies and it is not clear why. It was shown multiple times targeting of macrophages by FACS, which is more rigorous approach compared to IHC. Either authors should repeat experiments performed and published previously and discuss why pHLIP performance in their models (indeed strange models of tumor of 10-15 mm in diameter, very large tumors, which should be said in the abstract upfront) is different or provide rigorous justification and do not say that it will be subject of future studies. 

Authors referring actively in their response to PET studies with chelate-metal complexes, which is not an appropriate comparison. In the latest study it was clearly shown different distribution of metal-chelate PET and fluorescent pHLIP agents.

There is no explanation in the manuscript why pHLIP "targets" necrotic tissue. What is the mechanism (and we are actually ignoring fact that necrotic tissue uptakes everything very readily, no indication about that whatsoever). 

The suggestion to use pHLIP for delivery of alpha-emitter is at least strange, since kidney will be destroyed completely. 
